# What We Ask about When We Ask about Quarantine? Content and Sentiment Analysis on Online Help-Seeking Posts during COVID-19 on a Q&A Platform in China

**DOI:** 10.3390/ijerph20010780

**Published:** 2022-12-31

**Authors:** Luanying Li, Lin Hua, Fei Gao

**Affiliations:** 1Faculty of Social Sciences, University of Macau, Avenida da Universidade, Taipa, Macau SAR 999078, China; 2Faculty of Health Sciences, University of Macau, Avenida da Universidade, Taipa, Macau SAR 999078, China; 3Centre for Cognitive and Brain Sciences, University of Macau, Avenida da Universidade, Taipa, Macau SAR 999078, China; 4Institute of Modern Languages and Linguistics, Fudan University, Shanghai 200433, China

**Keywords:** COVID-19, Zhihu social platform, help-seeking post, LDA model, sentiment analysis

## Abstract

The COVID-19 outbreak, a recent major public health emergency, was the first national health crisis since China entered the era of mobile social media. In this context, the public posted many quarantine-related posts for help on social media. Most previous studies of social media during the pandemic focused only on people’s emotional needs, with less analysis of quarantine help-seeking content. Based on this situation, this study analyzed the relationship between the number of quarantine help-seeking posts and the number of new diagnoses at different time points in the pandemic using Zhihu, the most comprehensive topic discussion platform in China. It showed a positive correlation between the number of help-seeking posts and the pandemic’s severity. Given the diversity of people’s help-seeking content, this study used topic model analysis and sentiment analysis to explore the key content of people’s quarantine help-seeking posts during the pandemic. In light of the framework of uses and gratifications, we found that people posted the most questions in relation to help with information related to pandemic information and quarantine information. Interestingly, the study also found that the content of people’s quarantine posts during the pandemic was primarily negative in sentiment. This study can thus help the community understand the changes in people’s perceptions, attitudes, and concerns through their reactions to emergencies and then formulate relevant countermeasures to address pandemic control and information regulation, which will have implications for future responses to public health emergencies. Moreover, in terms of psychological aspects, it will help implement future mental health intervention strategies and better address the public’s psychological problems.

## 1. Introduction

Since December 2019, the COVID-19 pandemic has been spreading rapidly around the world, COVID-19 had led to more than 544 million confirmed cases and nearly 6 million deaths worldwide by 30 June 2022 (https://www.who.int/zh/emergencies-/diseases/novel-coronavirus-2019) (accessed on 13 July 2022). Due to the destructive and highly contagious nature of COVID-19, many countries have taken various measures to prevent the spread of the virus, such as quarantines and the lockdown of cities [1,2,3,4]. People are advised to keep socially distant or stay at home during the city’s lockdown [4]. Although these control measures are considered effective in stopping the spread of the pandemic, one still has to face the challenge of insufficient information or uncertainty [5]. Fortunately, compared to the 68 million Chinese Internet users during the SARS outbreak in 2003, the number of Chinese Internet users had increased to 989 million by 2021. With the development of the Internet, people are beginning to use social media to meet their social needs.

This study analyses the help-seeking posts made by quarantined people using social media in order to meet their needs, based on uses and gratifications theory. Uses and gratifications theory attests that individuals act on the desire to satisfy their needs and are increasingly involved in the production of media content, and it also suggests that individuals use mass media to satisfy their needs while instrumentalizing them in this direction [6,7]. In addition, uses and gratifications has been studied in the light of changes in different media, such as newspapers, radio, and television. With the development of new communication technologies, academic research has pointed to the role of social media in establishing a theory of uses and gratifications by applying the theory to understand the changes brought about by new communication technologies and online environments, such as how individuals use social media to meet their needs [8]. Specifically, many people used social media to obtain virus-related information by posts and retransmission when faced with a lack of awareness of the virus, which was always combined with psychological fear, confusion, and helplessness. To illustrate the performance of social media in the pandemic in China, several extant studies mainly focused on the search for online health information, the use of the Internet for self-care of chronic diseases, and the emotions of Wuhan netizens during the early stages of the outbreak [9,10,11]. Although these studies have increased our understanding of social media use during the pandemic by examining it from the perspectives of topic discovery, emotional tendencies, and remote support for disease patients, there are still some limitations. The existing research solely concentrates on one city or one time period in China, such as the city of Wuhan during the early outbreak phase of the pandemic. This is due to constraints in the scope of the studies, which have resulted in limited sample sizes. Second, there are restrictions on the subject matter of research, since it now either examines changes in emotional state over the course of a day from a micro perspective or focuses on the patterns of emotion distribution from a macro perspective. Finally, there is the limitation of the platform, as all of these studies use Weibo, the same social media site, to examine how social media works and produce some consistent findings. To highlight people’s online help-seeking behaviors during the pandemic and the contribution of social media in this process, the current study targets a specialized question-and-answer (Q&A) platform (i.e., Zhihu). Zhihu is an experience- and knowledge-sharing forum in the form of a Q&A format (similar to Quora) with more than 200 million users, with topics ranging across a variety of fields. In addition, Zhihu is a space for discussing popular topics, and its social nature allows for communication, sharing, and interaction, such that people’s requests for help on Zhihu can be a more accurate reflection of people’s mindsets and situations than on Weibo [12]. During COVID-19, Zhihu became not only a channel for people to follow the development of the pandemic and express their personal opinions but also an essential tool for Internet users who needed help with medical, policy, and health care [13]. Thus, the large amount of data generated reflects the state of information dissemination on the social platform and warrants systematic analysis.

In order to depict and analyze this phenomenon, this study explores the distribution of public demand for quarantine information by combining topic model analysis and sentiment analysis for quarantine-related help-seeking posts on Zhihu during the pandemic. Although topic model analysis has been employed in social media platforms, such as Twitter and YouTube, many countries have developed pandemic prevention strategies in accordance with their unique pandemic situations, and people’s social media usage is also varied. China has highly rigorous isolation prevention and control procedures. The daily lives and employment of individuals are greatly impacted by this strict policy. Social media shows their powerlessness, perplexity, and other emotional states. As a result, topic model analysis is used to investigate social media in China in this study. By exploring keyword distribution, sentiment distribution, and time distribution, this study could implicate a wide significance. Theoretically, it demonstrates that the uses and gratifications theory is still a useful framework for understanding the adoption and emergence of new communication media use practices, since the help-seeking posts on social platforms provide a window that allows for a good examination of this theory (such as receiving and seeking information and social interaction). Our study also extended the theory to activities taking place during significant public health events on the new social media platform. We gained a better understanding of the changes brought about by new communication technologies and the online environment through the implementation of the theory, which also helped to expand the theory’s applicability and content. In terms of the practical implications, firstly, identifying the relationship between people’s help-seeking posts about quarantine and the severity of the pandemic could provide insights into how people react to emergencies by using help-seeking posts as additional information for pandemic monitoring. Secondly, by examining the types of topics covered on social media, our findings would advance our understandings of people’s perceptions, attitudes, and concerns during the pandemic. Relevant departments can understand the different levels of impact of the pandemic on various aspects of society and use social platforms to better meet the needs of the public. Finally, by analyzing the emotional tendencies of the content of help-seeking posts, our findings will help with the implementation of future mental health intervention strategies and will also provide a reference for people to use social media to seek help in the future. 

## 2. Literature Review

The WHO defined help-seeking as the behavior of an individual who believes that he or she needs personal, psychological, and emotional assistance or health/social services to meet the need positively. Such a process involves acknowledging the need for help and taking practical action to receive treatment [14] (Table 1). During public crises, social media platforms become potential attention spaces for people to seek help. People are increasingly turning to cyberspace for social support in various areas, such as coping with everyday troubles and battling life-threatening illnesses [13,15,16]. Taken together, social platforms increase the opportunities to provide and receive social support from all sides. Specifically, social media is widely used for both help-seeking channel and health promotion when people are faced with difficult situations as it has a high update rate and can quickly disseminate information. As such, social media could allow the public, health organizations, and government agencies to engage, interact, and communicate about health issues [17]. 

In China, people’s habit of using social media to ask for assistance is a more common phenomenon. This is because the large population has made it challenging for people to access health care resources, including appointment scheduling, having short consultations, and having big knowledge gaps in health [18]. Online social platforms have become an important medium for most Chinese people to obtain information and express their emotions [19,20]. After the emergence of COVID-19, there have been increasing studies that touched upon the help-seeking posts related to the pandemic on social media in China. Currently, research combining the pandemic and social media in China focused on two aspects. The first line of research is concerned with the use of social media by people for remote online treatment during the pandemic. For example, Wu and Yu (2021) investigated the use of digital devices by older people for chronic disease management during and after the Wuhan city closure, where telemedicine provided a technological solution to alleviate people’s chronic disease management [9]. Another recent study discussed how social media was used for the self-care of chronic diseases and whether people with chronic diseases were satisfied with the outcome of social media online treatments, thus highlighting the information needs of vulnerable groups and the role of social media [21]. The research went into detail about how social media can assist chronically ill patients and the elderly remotely. The research’s results are constrained because it is focused on particular populations. The second category of research results focuses more on public opinion, psychological states, and emotional fluctuation. For example, from a macro perspective, based on the text emotion extraction method of emotion ontology, researchers discovered that a considerable rise in “anxiety” was caused by COVID-19’s rapid proliferation over a relatively short period of time [22]. Other researchers conducted in-depth analyses of relevant content on Weibo, created an event evolution map of online public opinion during the outbreak of the pandemic, and outlined the relevant guidelines for the causal evolution of public opinion using big data technology. They also studied the characteristics of social media topics and emotional changes from a time and space perspective [23]. These studies provide decision-making support and communication for macro-control coping strategies and risks through the analysis of media information, helping those departments accurately understand the subjective thoughts and feelings of the public. In addition, some scholars also conducted research from a micro perspective, focusing on the emotional changes of people in a certain city within a specific time. For example, Yu et al. (2021) analyzed data from a Chinese social media platform Weibo and found that moods changed over the course of a day during the COVID-19 pandemic [10]. Whereas all emotions were more prevalent in the afternoon and evening, fear and anger were more prominent in the morning and afternoon, and depression was more prominent at night [10]. Zheng et al. (2021) tracked changes in public mood in Wuhan during the first 12 weeks after the discovery of COVID-19 on the Weibo platform [11]. Their study found a progression from confusion and fear to disappointment/depression, depression/anxiety, and finally happiness/gratitude. This progression reflects the changing emotional energy of digital healthcare citizens and allows for intervention in future crises. 

In a nutshell, current help-seeking study is undertaken from the perspectives of remote help for disease patients, public opinion, and emotional tendencies. This advancement has broadened people’s knowledge of the usage of social media throughout the outbreak and can serve as a model for future emergencies. However, there are certain limitations to the study. First, the research scope constraints result in a small sample size, including location and time constraints. The current study solely focuses on a specific period in a particular Chinese city, which is the early outbreak of the pandemic in Wuhan city. The second limitation is the research platform’s limits. At the moment, the results data are concentrated on the Weibo platform, and the obtained results contain overlapping parts, such as the trend of emotional fluctuations. Finally, the research topic and substance have constraints. The current research focuses on the macro distribution of emotions or on small-scale emotional changes from a micro perspective. The research content is insufficiently comprehensive. In light of these limitations, this study used an experience- and knowledge-sharing question and answer platform with over 200 million users as the source of data extraction, which offers clear advantages over Weibo in testing help seeking behavior [12]. The association between the number of quarantine help-seeking posts and the number of new diagnoses at different time points in the data is examined not only from a macro perspective but it also examines, from a micro viewpoint, the emotional tendencies expressed in these posts regarding isolation, as well as the content that users prefer in their posts.

## 3. Method

### 3.1. Data Collection and Pre-Processing

In this study, we focused on Zhihu posts related to help-seeking about the quarantine of COVID-19. Using Zhihu application programming interfaces (APIs), we collected all original posts that contained the keywords of help-seeking (“求助”) and COVID-19 quarantine (“新冠疫情隔离”) between 1 January 2020 and 30 June 2022, which initially obtained 26,395 related posts. The following information was extracted: user ID, timestamp (the date and time of posting), and text (the help-seeking message posted by a user).

To eliminate noise in the obtained texts and improve the accuracy of word segmentation and further natural language processing (NLP) analysis, interfering information (i.e., HTTP hyperlinks, punctuations, spaces, and emoji codes) was firstly filtered out from original texts. Then, the concise texts consisting of less than five words were deleted. Additionally, we manually excluded posts that contained more than one missing piece of information. Finally, 10,685 Zhihu post entries with “the help-seeking of COVID-19 quarantine” were selected for further analysis (Figure 1).

### 3.2. Data Analysis

#### 3.2.1. Time Series Analysis

To identify the relationship between help-seeking posts about the quarantine of the pandemic and the severity of COVID-19, we firstly obtained the daily number of confirmed cases in China from JHU CSSE COVID-19 Data (https://github.com/CSSGISandData/COVID-19) (accessed on 1 January 2020). Then, we divided the number of Zhihu posts and combined the daily number of confirmed cases into different time points based on quarters from 2020 to 2022. Finally, the correlation between the number of Zhihu posts and the number of confirmed cases in 10 quarters from 2020 to 2022 were calculated using Python (Python Software Foundation: Wilmington, DE, USA).

#### 3.2.2. Topic Extraction and Classification Analysis

To investigate the topics posted on Zhihu for help-seeking posts related to quarantine during the pandemic, pre-processed Zhihu texts were hierarchically processed using the Latent Dirichlet Allocation (LDA) model [24,25]. The LDA model is an unsupervised machine learning method defined as a generative probabilistic model that provides semantic topics in large-scale document sets or corpora with probability distributions. In the LDA model, there are three layers (i.e., document–topic–word), in light of the notion that documents are represented by a multinomial distribution of topics, each characterized by a multinomial distribution of words [24]. The joint distribution of all variables in LDA model were showed in Equation (1) [26]. In general, LDA model is one of the most concerned models in the field of text analysis and has a wide range of applications for semantic annotation, dimensionality reduction, and text mining [27] than other text mining approaches, such as TF-IDF (term frequency–inverse document frequency), latent semantic analysis (LSA), or probabilistic LSA [26].
(1)$$p\left(\beta_{k},\theta_{D},z_{D},\omega_{D}|\alpha ,\eta \right)=\prod_{k=1}^{K}p\left(\beta_{k}|\eta \right)\prod_{d=1}^{D}p\left(\theta_{d}|\alpha \right)\prod_{n=1}^{N}p\left(z_{d,n}|\theta_{d}\right)p\left(\omega_{d,n}|z_{d,n},\beta_{d,k}\right)$$

$\omega_{d,n}$ is the only observable variable expressing the words in each document. $z_{d,n}$ is the dependent variable of $\theta_{d}$, and $\omega_{d,n}$ is the dependent variable of $z_{d,n}$, $\beta_{k}$. *n* is the total number of unique words in each document, and K is the number of topics. $\beta_{k}$ is the topic distribution for each document, $\theta_{d}$ is the topic distribution for each subject, and $z_{d,n}$ is each word given to the topics. η and α are the hyperparameters for the prior distribution of the per-document topic distributions $\theta_{d}$ and per-topic word distribution ($\beta_{k}$), respectively.

The texts we collected included many irrelevant words (i.e., “我” (I/me), “他/她” (he/she)), which may lead to confusion topic extraction. Therefore, before conducting LDA model, we performed word segmentation using a public stop word list (Baidu stop word). Word segmentation can give obvious separators between Chinese words and produce a more precise topic extraction. Then, the Jieba package in Python was adopted to perform the Chinese text segmentation.

Previous studies usually used the Genism package in Python, which contained the implementation of the LDA model [28]. In the current study, however, we employed the Mallet package, a java-based NLP processing algorithm package, which included an optimized and improved implementation of the LDA model [29,30]. Compared with the original LDA model in the Genism package, the Mallet topic model includes a speedy and highly scalable implementation of Gibbs sampling, which thus implicates an efficient document topic hyperparameter optimization method.

The key inputs in the LDA algorithm were two hyperparameters, η and α, and the number of topics, K. The larger values of the hyperparameters η and α result in more even distributions, whereas smaller values lead to more concentrated distributions over topics or words. As suggested from Mallet package tutorial, a value of 1/topic numbers can lead to more meaningful outcomes and were selected in current LDA model. The optimal number of topics K was initially determined through repeated experiments to be 20. The LDA model’s topic-terminology lists contained the vocabulary of each initial topic and their frequency. The document–topic lists indicated the likelihood that each Zhihu text was associated with each of the first 20 topics. In terms of the determination of the optimal number of topics, coherence scores were taken into consideration. By merging similar topics and discarding irrelevant topics, twenty topics were reduced to six topics. In fact, a help-seeking post may be included in more than one topic due to the words it contains. However, using LDA analysis, each post was only presented the topic with the highest probability. In current study, T-SNE (t-distributed stochastic neighbor embedding) analysis was used to visualize and give a clear distribution of all help-seeking posts [31]. Furthermore, pyLDAvis package in Python was used to visualize the topic extraction results, with lists of the 30 most salient words in each topic. Finally, we extracted 10 representative words in each topic to manually label the topics.

#### 3.2.3. Sentiment Analysis

To analyze the sentiment tendency of the help-seeking posts related to pandemic quarantine, sentiment analysis was conducted on the texts after word segmentation by using the Snownlp package in Python. Only adjectives which appeared in more than 0.1% of the texts were kept in the pre-processed Zhihu texts for further analysis. Through the Bayes model in the Snownlp package, we constructed a classification model to predict the sentiment strength of an adjective in Zhuhu texts. The occurrence probability of positive or negative sentiment was used to determine the sentiment strength of an adjective [32].
(2)$$P\left(Class|Adjective\right)=\frac{P(Adjective|Class)P\left(Class\right)}{P\left(Adjectve\right)}$$

Then, a common linear transformation was used to normalize sentiment strength coefficients for the range from 0 to 1. If the sentiment tendency was positive, the result would be close to 1. In contrast, the negative sentiment tendency would lead the result to approximately 0 [33]. Finally, the top 10 frequent adjective words were extracted.

## 4. Results

### 4.1. Post Description

By searching for and pre-processing the data of help-seeking posts about the pandemic quarantine on the Q&A platform Zhihu, we collected a total of 10,685 posts related to the given keywords. The word cloud (Figure 2) and the top 50 frequent words (Table 2; also see Table A1 for English and Chinese bilingual) represented the key words of the most concerned questions, such as “核酸” (nucleic acid), “电话” (phone number), and “物资” (goods and materials). These keywords reflected the massive difference in the public’s interest in different types of help-seeking information needs, including local COVID-19 outbreak, prevention measures, and hospital. Additionally, the number of Zhihu posts was divided into ten phases based on quarters from 2020 to 2022. We obtained the phase distribution of help-seeking information needs and new confirmed cases according to the ten phases. The results indicated that the pandemic’s help-seeking information needs were intense in the first quarters (from 1 January 2020 to 31 March 2020). After that, help-seeking information needs declined from 1 April 2020 to 31 December 2021. Yet, help-seeking information needs increased significantly from 1 January 2022 to 30 June 2022 (Figure 3a). In particular, the number of Zhihu posts was positively correlated with new confirmed cases during the pandemic, *R*^2^ = 0.6554, *p* = 0.0045 (Figure 3b).

### 4.2. Topic Modelling

To obtain the best LDA model performance in predicting the samples, the number of topics for LDA model training was set from one to twenty with a step of one. An increasing number of topics set for model training could result in a better performance for prediction, while excessive classification for topics may lead to model overfitting issues. Therefore, coherence scores were used to compare different model settings and decide the optimal number of topics. As shown in Figure 4, a peak was identified with the coherence scores when the number of topics for model training is set as six.

From the selected LDA model mentioned above, this study obtained important words related to the corpus, and the sampling period was from January 2020 to June 2022. These keywords can better represent the overall picture of the topics that were of interest to the public. Multidimensional distance for the six topics (Figure 5) showed the representative distance of each topic, although there were overlapping areas in 5th topic and 6th topic, which were primarily related to the help information of the prevention measures. Based on the LDA algorithm we performed, the top 30 most salient words of the selected topics can be initially obtained. After clustering the salient terms according to the manual labels, 10 representative words for one topic were selected from the top 30 most salient words (Table 3; also see Table A2 for English and Chinese bilingual). The six discovered topics, each with 10 representative words, were *quarantine assistance* (25% in whole Zhihu posts), *quarantine location* (23%), *people* (16%), *Epidemic treatment* (14%), *Epidemic prevention* (13%), and *Government information* (9%) (Table 4; also see Table A2 for English and Chinese bilingual). This finding could suggest the most concerned topics for pandemic quarantine, among which five topics were about help with information related to the pandemic quarantine measures and quarantine assistant information for different types of people (i.e., *quarantine assistance*, *quarantine location*, *people and government information*). Representative questions focused on the solutions available in different types of people for “*how long home quarantine will last*” and “*how to deal with limited supplies*”. In addition, two topics were directly related to the pandemic outbreak (i.e., *Epidemic treatment* and *Epidemic prevention*). Interestingly, for the topic of *quarantine assistance*, psychological help was one of the major questions, as manifested by the keywords of *emotion*, *psychology*, *depression*, *anxiety*, among others. In response to these questions, the Zhihu platform can accurately recommend people with expertise in different fields, thus effectively solving the problem of uncertainty in the process of information dissemination.

### 4.3. Sentiment Tendency Results

Based on sentiment prediction results, the sentiment tendency of Zhihu texts was divided into three categories: positive (values ranging from 0.6 to 1.0), neutral (0.4 to 0.6), and negative (0.0 to 0.4). Statistical results (Table 5) showed that help-seeking posts about the quarantine of the pandemic in Zhihu from January 1, 2020 to June 30, 2022 expressed more negative and dissatisfaction sentiment (*n* = 4639) with a low emotional tendency value (0.112 ± 0.188). Some Zhihu help-seeking posts (*n* = 4473) expressed objective content obtaining a medium emotional tendency value (0.516 ± 0.134). Only several Zhihu help-seeking posts (*n* = 1573) expressed good expectations holding a higher emotional tendency value (0.692 ± 0.097). These findings indicated that the model prediction results aligned with public cognition.

To further understand the frequency of specific keywords corresponding to the sentiment analysis results, the top 10 keywords contributing to sentiment prediction were listed in Table 6 (see Table A3 for English and Chinese bilingual). The results only presented two keywords with a positive sentiment, mainly *fighting* and *healthy.* By contrast, there were eight negative keywords, including *depressed*, *worried*, *repressive*, *severe*, *anxious*, *corrupted*, *angry,* and *helpless*. The top 10 emotional keywords were primarily negative, with only two positive sentiments at a relatively lower rank. This showed that during COVID-19, help-seeking posts about the quarantine pandemic in Zhihu from 1 January 2020 to 30 June 2022, mainly expressed negative sentiments.

## 5. Discussion

### 5.1. Time Series Discussion

Through time series analysis, this study examined 10,685 posts about quarantine help-seeking during the pandemic and obtained the following findings. Firstly, people’s concerns about the pandemic changed from 1 January 2020 to 31 March 2020, when the demand for information reached a high level during the pandemic, with a relatively higher demand for information seeking help. From 1 April 2020 to 31 December 2021, the demand for information slowly contracted from the high level and remains stable, entering a plateau period. Eventually, it entered a volatile period again, with demand for help picking up and the pandemic regaining a high level of public attention. This finding is consistent with the increased activity of people on social networking sites following a pandemic outbreak [34]. Other scholars have obtained similar findings across different disaster events. Michael et al. concluded that trends in dengue disease outbreak events were highly correlated with trends in the number of Weibo texts [35]. However, the scope of this study is more precise and shows more accurately that the number of posts on Zhihu is positively correlated with the number of new cases diagnosed during the outbreak. This shows that public social media postings are not random and that people are more inclined to seek help after being diagnosed. This result is in line with the uses and gratifications theory, where audiences are targeted, purposeful, and motivated in their media choices and use, and they want to use them to satisfy needs based on psychological or social needs [36]. People affected by the pandemic have even more psychological and varying needs that need to be satisfied through social media. In other words, people’s help-seeking behavior during a pandemic is influenced by the development of the pandemic. Applying this theory to help-seeking posts during quarantine also extends the scope of the study of the theory. The conclusions drawn from the analysis in this section allow for a more objective understanding of people’s responses to emergencies. In the future, help-seeking could also be used as additional information for epidemiological surveillance.

### 5.2. Topic Discussion

On Zhihu, people posting for help come from different regions, professions, and ages, so the content of the posts is original and diverse. Our analysis showed that there was an overlap between the results of the content of the help-seeking posts initiated on Zhihu and previous research. For example, the results of these studies show that people are generally concerned about prevention policies and the impact on their lives and work. However, the data from the previous studies were relatively limited as they only focused on health needs or concentrated on the pandemic’s early stages [28,37]. Our study covered a broader range of data, for instance, the LDA model’s results show six categories of topics for which people ask for the most help, along with the top ten words of concern for each category. Of these six categories, five discussed with information about pandemic quarantine measures and information about quarantine assistants for various types of people (such as information about quarantine assistance, quarantine locations, people, and government information). In addition, people pay the highest attention to the quality assistance and quality location. There are more than 2000 posts, which represent 25% and 23% of the total. The words “workplace”, “living at home”, “movement”, “goods”, and “materials” are used. The representative concerns focus on the potential solutions for various types of people, such as “how long will the family quarantine last” and “how to deal with the limited supply”. This demonstrates that people must consider how the isolation site will affect their ability to work and attend school during that time and that they are also quite worried about problems related to their daily lives, such as how the government will provide them with food and other supplies. This result supports the notion in uses and gratifications theory that audiences are rational, understand their interests and motivations, and are able to articulate them clearly. The researcher is able to infer the purpose of the audience’s use of the media based on the audience’s reactions [38]. By studying the types of topics on social media, we can gain a deeper and more comprehensive understanding of the impact of the pandemic on people in various aspects and also a good understanding in people’s perceptions, attitudes and concerns about the pandemic during the process. This information can be used as a reference and guide for the relevant departments to formulate and launch relevant response measures to address various aspects of pandemic control, protection of people’s rights and interests, and information regulation, as well as a reference for future responses to public health emergencies.

### 5.3. Sentiment Tendency Analysis

By analyzing the emotional tendencies of the post content, we found that the public’s emotional tendencies were on the negative side. The previous study looked at 727 posts over a nine-month period and concluded that the predominant emotion was fear [13]. Yet, our sentiment analysis showed that the dominant emotion was “depressed” during the pandemic. This discrepancy might result from the differing sampling time. Luo et al.’s data came from the early stages of the pandemic when people did not know enough about quarantine and COVID-19, and the vaccine was not widely available, which might profoundly contribute to people’s uncertainty and fear. In contrast, this study included more recent posts over a longer time, which could be more reflective of people’s internal state. In other words, when people have obtained more adequate and more scientific information about the outbreak, which could largely drive away the unnecessary fear towards the illness. Instead, people may feel more depressed due to the increased number of quarantine lives caused by the continued recurrence of the pandemic, in addition to the negative emotional bias of people living in quarantine, even those with chronic or psychological illnesses who cannot go to hospital and have to resort to social media. The results of previous studies have demonstrated that people are more likely to seek help on social media during the pandemic, mostly from patients for their illnesses, such as how to help vulnerable groups with remote treatment during the pandemic and the impact on patients with chronic diseases [39,40]. There are also studies that focused on emotional aspects [17] and the emotional state of the population confined to a particular city in China [34]. The current study further ensured that the diverse sources of data did not place restrictions on cities, which is a new breakthrough in psychological research, and these results also confirm that people choose media according to their needs and that media is considered to be a helpful source of treatment for people’s emotions, contributing to the implementation of future mental health intervention strategies [38]. Specifically, it will help the relevant authorities to grasp the emotional trends of the public on the Internet so that they can provide targeted professional answers and responses, provide more knowledge on mental health management and regulation to the public, and emphasize the role of emotional monitoring and emotional feedback, alleviating public anxiety and panic, and thus better addressing people’s psychological problems during the pandemic.

### 5.4. Limitation and Future Direction

Our research has shown the usefulness of using computational methods to explore help-seeking during the pandemic. For example, the vast amount of content on social media and the changes in people’s emotions during the pandemic would be difficult to capture by using qualitative methods. However, some caveats of the current study still need to be noted. Firstly, for data processing, manual judgments were performed to improve the accuracy of sentiment analysis, so the experimental results’ objectivity needed to be considered in further study. Secondly, only minor optimizations were modified to the LDA topic model and, as such, based on more specific topics and comments on the Zhihu platform, further research can be devoted to combining more algorithms and proposing a more specific topic model for the Zhihu platform. Finally, although the data on the Zhihu platform can provide insight into the state of the public’s need for help, the data restricted to China or the sample size of one platform is rather limited. In light of the aforementioned circumstance, data for future research can be gathered from various social media platforms, and data from other nations can be compared for a more comprehensive analysis.

## 6. Conclusions

In this study, we examined the public’s quarantine-related help-seeking posts on an online Q&A platform during the COVID-19 pandemic. In general, attitudes and confusion about quarantine were reflected in social media during the outbreak, and people used social media as an important source of information and help-seeking. We found that the spread of the disease had an impact on how people reacted while seeking help during the pandemic. A positive correlation exists between the quantity of posts on Zhihu and the number of new diagnosed cases during the outbreak. The study results also listed the top six concerns people have while seeking help. According to the results, individuals are most concerned about quarantine assistance and quarantine locations, and there were negative emotional tensions among the public. This study contributed to both theoretical and practical research in several ways. In a word, this study contributes to expanding the scope and content of the theory. In terms of practical implications, it would also be beneficial for the government to comprehend how people react to public health emergencies, though these results may be utilized as a guide when developing relevant policies in the event of similar emergencies in the future.

## Figures and Tables

**Figure 1 ijerph-20-00780-f001:**
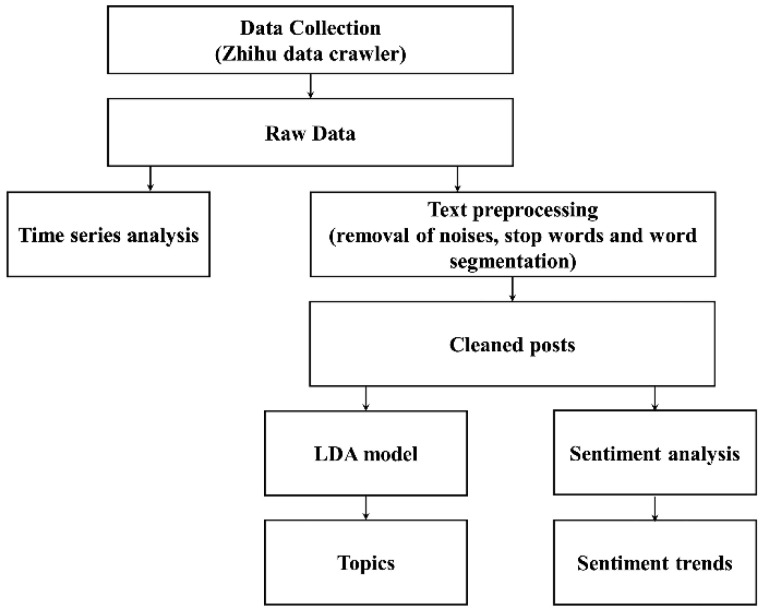
Framework of multi-analysis for Zhihu posts.

**Figure 2 ijerph-20-00780-f002:**
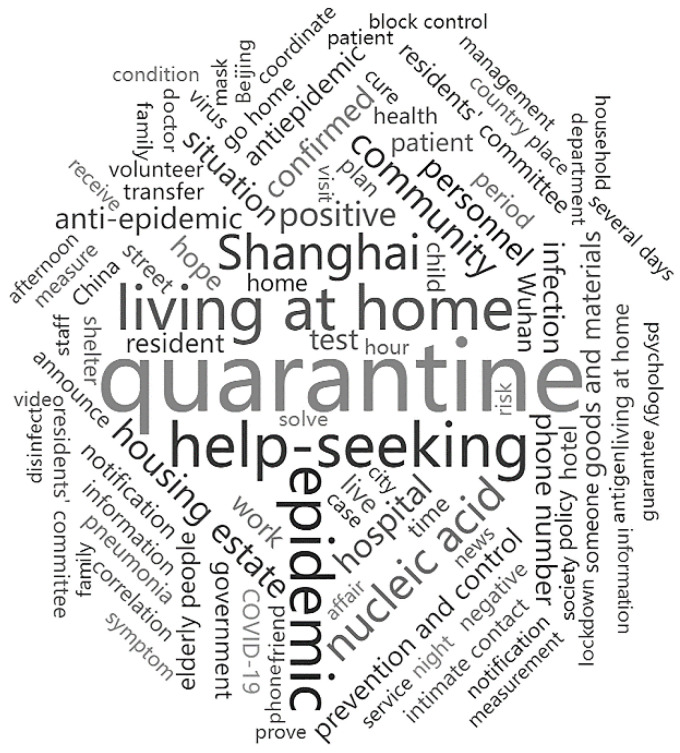
Word cloud of top 100 terms in Zhihu posts.

**Figure 3 ijerph-20-00780-f003:**
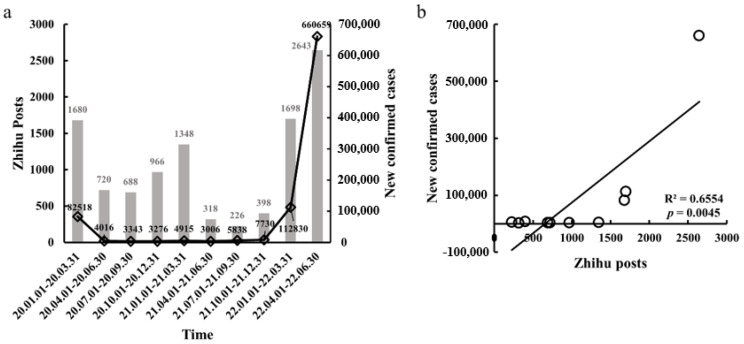
Basic description of Zhihu posts and its correlation with new confirmed cases. (**a**) The number of newly diagnosed cases and posts on Zhihu from 2020.01.01–2022.06.30. The gray bar graph shows the number of quarantine posts in different quarters, and the black hollow rhombus shows the number of newly diagnosed cases in different quarters. (**b**) Relationship between the number of newly diagnosed cases and the number of quarantine posts in Zhihu during the pandemic.

**Figure 4 ijerph-20-00780-f004:**
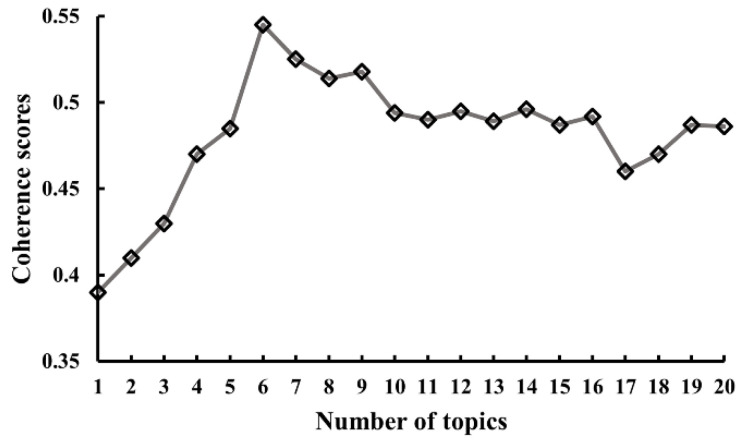
The changes in coherence scores of LDA model using different settings of number of topics.

**Figure 5 ijerph-20-00780-f005:**
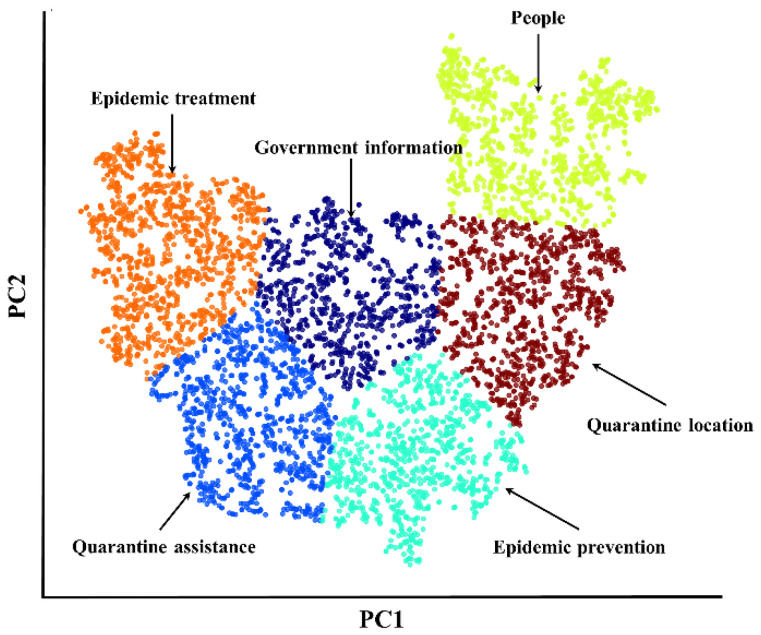
Visualization of LDA model (the number of topics is 6). Multidimensional distance scales provided a visual representation that projects the distribution among 6 topics.

**Table 1 ijerph-20-00780-t001:** Literature review.

Author	Objective of study	Method	Findings
[14] Castonguay et al., 2016	Exploring the help-seeking process	Interviews were used to examine the health belief model	The main barrier preventing help-seeking was fear of the unknown treatment process.
[15] Alshaabi et al., 2021	Enhancing any analysis that may be useful during the pandemic, as well as retrospective surveys.	Multiple languages and *n*-gram analysis	In all languages, the word ‘virus’ peaked in January 2020, followed by a decline in February, and then a surge in March and April. The world’s collective attention declined as the virus spread out from China.
[16] Andersson and Sundin, 2021	Identification of theoretical perspectives relevant to the analysis of mobile media practices and discussion of the ethical implications of these perspectives.	Theories of people’s behavior at the scene of trauma were combined with discussions about witnessing both in and through the media.	Mobile bystanders must be considered simultaneously as transgressors of social norms and as emphatic witnesses behaving in accordance with the digital media age
[13] Luo et al., 2020	Exploring the driving force behind the retweeting of online help-seeking posts.	An analytical framework that emphasizes content features was used	The importance of individual information completeness, high proximity, instrumental support seeking.
[17] Thackeray et al., 2012	Assessing the most common social media used by state public health departments (SHDs) and the frequency with which social media is used to interact to engage audiences.	Cross sectional study	A total of 86.7% had a Twitter account, 56% a Facebook account, and 43% a YouTube channel. SHDs had very little interaction with audiences. The most common topics for posts and tweets related to staying healthy and diseases.
[18] Shi and Kim, 2019	Examining the factors that lead young Singaporeans to seek advice and adopt a self-help approach.	Risk perception attitude framework and theory of planned behavior were integrated.	The nature of focal behavior and attitudes are boundary conditions of the interaction effect between perceived risk and efficacy.
[19] James and Cedric Harville, 2016	Assessing the relationship between electronic health literacy (EHL) and willingness to participate in mobile health (mHealth) research.	Questionnaire	Significantly higher eHEALS scores among women, smartphone owners, those who use the Internet to seek health information, and those willing to participate in mHealth research.
[20] Ji, 2013	Exploring the help-seeking behavior of university students when experiencing psychological distress.	Questionnaires and structural equation modeling	Students with positive help-seeking intentions, attitudes, and more indirect help-seeking experiences are more likely to engage in help-seeking behavior.
[9] Wu and Yu, 2021	Exploring the willingness of older people in Wuhan to use digital devices for chronic disease management during and after the city lockdown.	Semi-structured interviews	Cultural background, medical knowledge, and socio-economic infrastructure play a key role in influencing the perception and use of remote medical and remote care services by older people.
[21] Kor et al., 2021	Satisfaction with COVID-19-related online information among patients with and without chronic conditions.	Online survey	The majority of PWCD who looked to social media for online information related to COVID-19 had significantly lower levels of information satisfaction than those without chronic health conditions.
[22] Shi et al., 2022	Analyzing the development of online public opinion in terms of fine-grained emotions during the COVID-19 outbreak in China	LDA model and sentiment analysis	A strong emotional impact is observed during holidays. Central cities reacted more strongly to the COVID-19 outbreak than surrounding cities.
[23] Zhu et al., 2020	Exploring social media topics and shifting sentiment features.	LDA model	Discovered the new characteristics of the “double peaks” of public opinion. Popular topics have the characteristic of slowly declining over time.
[10] Yu et al., 2021	Exploring the significant events that influenced emotional changes during the COVID-19 pandemic	Sentiment analysis	Negative emotions were the most salient emotions detected on Weibo during the night.
[11] Zheng et al., 2021	Investigating the evolution of public sentiment in Wuhan, China in the first 12 weeks after the emergence of COVID-19	Longitude analysis	The study found a progression from confusion/fear to disappointment/frustration, to depression/anxiety, finally to happiness/gratitude.
[12] Zhang et al., 2021	Exploring how to properly monitor We Media and effectively manage its violations	A tripartite evolutionary game model of government, We Media, and public participation was constructed	Government regulation plays an important role in restricting We Media’s information release

**Table 2 ijerph-20-00780-t002:** Top 50 frequent words in Zhihu posts.

No.	Term	Frequency	No.	Term	Frequency	No.	Term	Frequency
1	quarantine	16,881	18	work	2704	35	home	1640
2	help-seeking	10,672	19	prevention and control	2572	36	plan	1624
3	living at home	10,536	20	hope	2548	37	COVID-19	1604
4	epidemic	10,128	21	resident	2496	38	transfer	1600
5	Shanghai	7556	22	live	2460	39	health	1540
6	nucleic acid	6640	23	goods and materials	2408	40	go home	1528
7	community	4964	24	patient	2240	41	information	1528
8	housing estate	4428	25	government	1988	42	pneumonia	1520
9	hospital	4084	26	Wuhan	1960	43	shelter	1476
10	positive	3972	27	elderly people	1896	44	negative	1456
11	personnel	3468	28	Residents’ committee	1844	45	policy	1432
12	situation	3424	29	notification	1844	46	announce	1432
13	confirmed	3388	30	period	1820	47	intimate contact	1408
14	anti-epidemic	3164	31	time	1768	48	China	1364
15	test	3072	32	hotel	1768	49	solve	1352
16	phone number	2836	33	child	1724	50	psychology	1324
17	infection	2820	34	street	1704			

**Table 3 ijerph-20-00780-t003:** Ten representative words of each topic and the results of topic coding.

Topic No.	Label	10 Representative Words Selected from the Top 30 Most Salient Words
**1**	Quarantine assistance	phone number, information, notification, service, goods and materials, live, visit, psychology, emotion, help-seeking
**2**	Quarantine location	living at home, company, housing estate, hospital, hotel, shelter, living at home, home, workplace, school
**3**	People	patient, elderly people, child, friend, family, volunteer, mother, in person, medical personnel, doctor
**4**	Epidemic treatment	positive, confirmed, quarantine, transfer, cure, antigen, intimate contact, solve, coordinate, infection
**5**	Epidemic prevention	epidemic, nucleic acid, mask, measurement, lockdown, block control, protection, inspect, risk, disinfect
**6**	Government information	China, anti-epidemic, hope, government, residents’ committee, announce, news, management, report, center for disease control and prevention

**Table 4 ijerph-20-00780-t004:** Description of the six topics.

Topic No.	Keywords	Posts Number	Posts Percentage
**1**	phone number, information, notification, service, goods and materials, live, visit, psychology, emotion, help-seeking	2671	0.25
**2**	living at home, company, housing estate, hospital, hotel, shelter, living at home, home, workplace, school	2458	0.23
**3**	patient, elderly people, child, friend, family, volunteer, mother, in person, medical personnel, doctor	1710	0.16
**4**	positive, confirmed, quarantine, transfer, cure, antigen, intimate contact, solve, coordinate, infection	1496	0.14
**5**	epidemic, nucleic acid, mask, measurement, lockdown, block control, protection, inspect, risk, disinfect	1389	0.13
**6**	China, anti-epidemic, hope, government, residents’ committee, announce, news, management, report, center for disease control and prevention	962	0.09

**Table 5 ijerph-20-00780-t005:** Emotional tendency in sentiment analysis.

	Mean	Standard Deviation	Frequency
**Positive**	0.692	0.097	1573
**Neutral**	0.516	0.134	4473
**Negative**	0.112	0.188	4639

**Table 6 ijerph-20-00780-t006:** Top 10 adjective keywords in sentiment analysis.

Key Words (Adj)	Emotional Polarity	Frequency
Depressed	Negative	874
Worried	Negative	786
Repressive	Negative	655
Severe	Negative	421
Anxious	Negative	329
Corrupted	Negative	194
Fighting	Positive	161
Healthy	Positive	86
Angry	Negative	27
Helpless	Negative	14

## Data Availability

Data and code are available from the authors on request.

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
