# Peer review of "What We Ask about When We Ask about Quarantine? Content and Sentiment Analysis on Online Help-Seeking Posts during COVID-19 on a Q&A Platform in China"

_ijerph, 2022, doi:10.3390/ijerph20010780_

Round 1

Reviewer 1 Report

The article deals with an important and timely subject: Online Help-seeking Posts during COVID-19. I enjoy reading the paper. However, it is not comprehensive and deep enough. Providing some suggestions/inquires for your reference:

1.     The introduction and literature review are lengthy and out of focus. Narrative makes up most of the two sections. What are the research motivations? what are the research gaps in previous literature? What are the research questions?

2.     Data analysis and discussions are not novel, methods and findings are expecting.

3.     What are the paper's primary contributions? The discussion of theoretical and practical implications are rather limited.

Overall, the topic is interesting. However, because of inadequate preparation for the theory and lack of the thorough study, problems exist both in theory and in practice.

Reviewer 2 Report

Nice work has been prepared, but it is very inadequate at some points. I will briefly state these points.

The Introduction section needs to be shorter and more academic language. In this section, academic contributions and gaps in the literature should be mentioned. Thousands of studies were prepared with sentiment analysis and text mining methods using Twitter data. What is the difference in this study? What is academic and practical contribution?

Please write this by referencing academic sources in the introduction section.

It is unnecessary to talk about social media channels in China constantly. In the method section, it is sufficient to the state from which platform and how the data were obtained. The ability to generalize results in academic studies is an important feature. It is not correct to constantly emphasize that it is made in China. This is a research limitation and an important obstacle to evaluating the results. This should be stated in the research constraints. 

Linking to a web page within a page is incorrect (page 2, lines 60,53, etc.). Any additional information you show in parentheses makes the article difficult to read.

The literature review needs to be more comprehensive. Prepare a more detailed literature review and include it in your article in table form.

It is not correct to refer to the concept of LDA in the following sentence. 

"To investigate the topics posted on Zhihu for help-seeking posts related to quarantine 258 during the epidemic, pre-processed Zhihu texts were hierarchically processed using the 259 Latent Dirichlet Allocation (LDA) model (Samaras et al., 2020; Shen et al., 2020)"

I have indicated the resources you need to show for the LDA method. Fix reference errors like this. Always show the original, real source.

1- https://www.jmlr.org/papers/volume3/blei03a/blei03a.pdf - Latent dirichlet allocation DM Blei, AY Ng, MI Jordan Journal of machine Learning research 3 (Jan), 993-1022

2- https://oar.princeton.edu/bitstream/88435/pr1bv3w/1/OA_IntroductionProbabilisticTopicModels.pdf - Probabilistic topic models

Not Samantas or Shen

Include text mining studies prepared using the LDA method in your article. Why was the LDA method chosen? The studies below are some examples;

1- https://doi.org/10.1016/j.jairtraman.2019.101760 (Emotion Analysis and LDA Example)

2- https://doi.org/10.1016/j.eswa.2019.03.001 (LDA example, Text Mining)

3- https://doi.org/10.1177/03611981221112096 (the literature review should be shown in table form as in this research. The parameters and visualization used in the LDA method can be used)

4- https://doi.org/10.1016/j.eswa.2017.03.020 (LDA method)

You want to publish your paper in an international journal, so show Figure 2. Word cloud in English.

Figure 3 needs to be explained. It must be prepared again. The text is nested. Please have consistency in the figures you prepare throughout the article. If possible, use dark tones such as black and gray rather than red, green, yellow, or blue.

It is not appropriate to give Chinese and English words together. It makes it difficult to focus. Include only English words in the article. Chinese words can also be given in the appendices.

Why are the parameters used in the LDA method not included? Please visualize your topics with the t-Sne method so the reader can understand the distribution more clearly.

The discussion section must be better. This section shows the importance of research and concrete outputs !

This section should have the following headings with more details :

1- Academic Implementation
2- Practical Implications
3- Limitation and Future Studies

Good Luck

Round 2

Reviewer 1 Report

My recommendation at the time was to accept the manuscript in its original form unless the authors saw scope in making minor amendments. It is a pleasure to see that my original positive comments stand for this newly-submitted version, and that my observations about potential improvements have also been addressed successfully.

Reviewer 2 Report

The authors made a great effort and prepared revisions in a short period. Congratulations.